# Quantities and Units in Chemical and Environmental Engineering

Peter Glavič 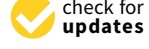

Department of Chemistry and Chemical Engineering, University of Maribor, Smetanova 17,
SI-2000 Maribor, Slovenia; peter.glavic@um.si

**Abstract:** The International System of Quantities (ISQ) shall be used in education and textbooks, in scientific and engineering journals, in conference papers and proceedings, in industry, among others. The names of quantities together with their symbols and units are being published by the International Organization for Standardization, the standard ISO 80000 Quantities and units, composed of 13 parts. Mathematics and natural sciences (physics, light and radiation, acoustics, physical chemistry, atomic and nuclear physics, condensed matter physics) compose most of the parts. In addition, some engineering disciplines (mechanics, thermodynamics, electromagnetism) and characteristic numbers are covered. The units are based on the International System of Units (SI). Unfortunately, chemical and process engineering, as well as environmental engineering and engineering economics, are not dealt with in the standard. In this paper, they are proposed as an additional part of the ISO standard with a tentative name Chemical and environmental engineering. The additional part of the standard is suggested to include (a) reaction and separation engineering together with mass transfer and reaction kinetics, (b) process design, control, and optimization, (c) process economics, mathematical modeling, operational research, and (d) environmental engineering with climate change, pollution abatement, an increase in resource efficiency, zero waste and circular economy. The number of quantities is planned but not limited to about 70, the average of ISO 80000 parts. Each quantity item contains a quantity name and definition (including an equation if suitable), SI unit, and remarks (running number will be added later). The rules are defined in ISO 80000-1 General rules, and the practice of the other ISO 80000 parts is respected; the quantities already included in the other parts are not repeated. In addition, the IUPAC (International Union of Pure and Applied Chemistry) Green Book rules are respected. The literature used included traditional textbooks, encyclopedias, handbooks from the chemical engineering and environmental fields. Some common mistakes in printing symbols of quantities and units are mentioned.

**Keywords:** chemical; environmental; engineering; quantities; standard; symbols; units

## 1. Introduction

After the letters, numbers, and units, the system of quantities was the last one developed and standardized globally. Quantity is a property of a phenomenon, body, or substance, where the property has a magnitude that can be expressed by means of a number and a reference (unit). The first international organizations trying to standardize chemical and physical quantities have been the International Union of Pure and Applied Physics, IUPAP (established in 1922), and the International Union of Pure and Applied Chemistry, IUPAC (formed in 1919). IUPAP prepared its first edition of Symbols, Units and Nomenclature in Physics in 1961 for official use only; its 1987 revision is available online [1]. IUPAC published the first edition of the Manual of Symbols and Terminology for Physicochemical Quantities and Units in 1969 [2]. After the 3rd edition, they changed the title of the manual and published it as a Green Book, again with three editions [3]. The Green Book is available on Internet, too.

In 1988, the International Organization for Standardization, ISO, in cooperation with International Electrotechnical Commission, IEC, published the first edition of international

standard ISO 31 Quantities and units in 13 parts [4], and ISO 1000 SI units and recommendations for use [5]. In 1992 a new version of both standards was published. In 2009, the two standards were substituted by ISO 80000 Quantities and units containing 13 somewhat reorganized parts [6]; the last edition was published in 2019, the exception being the parts mentioned in parentheses (the parts 1 and 6 are planned to be updated in the year 2021):

(1)   General (2009)
(2)   Mathematics
(3)   Space and time
(4)   Mechanics
(5)   Thermodynamics
(6)   Electromagnetism (2008)
(7)   Light and radiation
(8)   Acoustics (2020)
(9)   Physical chemistry and molecular physics
(10)   Atomic and nuclear physics
(11)   Characteristic numbers
(12)   Condensed matter physics
(13)   Information science and technology (2008)

The general part contains information about quantities and units, printing rules, rules for terms in names for physical quantities, rounding of numbers, and logarithmic quantities. In parts 3–13 (part 14 has been withdrawn), the quantities of each subset are listed, including item number, quantity name, symbol and definition, unit symbol, and eventual remarks. Since 2019, each part has an alphabetical index of quantities at the end to enable searching for the items. Three further parts, 15–17 (Logarithmic and related quantities, Printing and writing rules, Time dependency), are under development [7].

As can be seen from the above-cited list of ISO 80000 parts, chemical and process industries (CPI) are not included, although they are very important in many respects (turnover, profit, investments, employment, research, etc.). Besides the chemical industry, CPI involves pharmaceutical, cellulose and paper, metal, ceramic, textile, food and beverage, and other industries. The area includes process, plant, and equipment modeling, design, construction, analysis, optimization, operation, control, process economics, safety, hazard assessment, transport phenomena, etc.

There is also no standard on quantities in environmental science and engineering, although we are in the climate change/crisis, facing species extinction, pollution, and raw-materials scarcity. Even quantities in the ISO 14000 family on environmental management, e.g., the performance indicators in ISO 14031 standard, are not obeying the ISO 80000 and SI rules [8]. The Paris agreement, European Green Deal, Net-zero emissions by 2050, sustainable development goals are some of the most frequent buzzwords that we are facing every day. They are dealing with greenhouse gas emissions, renewable sources, critical raw materials, biodiversity, resource efficiency, zero waste, circular economy, etc. Therefore, it is necessary to define internationally agreed names, symbols, and units for the quantities used in the area.

## 2. Methods

The literature search included chemical and environmental engineering textbooks, manuals, standards, lexicons, encyclopedias, and handbooks, e.g., Ullmann's Encyclopaedia [9], Perry's Chemical Engineers' Handbook [10], SI brochure [11], and Google searches. As the number of items is limited, the most important quantities have been selected according to the importance and frequency of their usage using the cited references and personal experience.

The proposal starts with chemical engineering quantities, continues with process economic ones in design, and finishes with the environmental ones. Some common mistakes in symbols of quantities and units are mentioned. They can also be found in the literature [12].

The proposed terms will be discussed at some professional meetings and published in this journal. After improvements, they will be sent to the EFCE (European Federation of Chemical Engineering), the AIChE (American Institute of Chemical Engineers), the IChemE (British Institution of Chemical Engineers), and the DECHEMA (Deutsche Gesellschaft für chemisches Apparatewesen). After their approval, they will be asked to send the proposal to the Technical Committee ISO/TC 12 Quantities and units.

## 3. Results and Discussion

Chemical and process engineering quantities are very exhaustive as they cover many topics based on chemistry, physics, mathematics, economics, etc. that deal with a very broad range of materials, methods, and equipment, e.g., ([9,10]):

- Principles of fluid and particle dynamics, heat and mass transfer, chemical thermodynamics and kinetics, statistics, and optimization methods;
- Very diverse reactions: homogeneous liquid or gas ones, gas-liquid, or gas-liquid-solid ones, using blast, or rotary furnaces, fixed or fluidized beds, heterogeneous gas catalysis, electrolysis, photo-, or plasma-chemistry, etc.;
- Unit operations such as size reduction and classification, transportation, and storage, mechanical, magnetic, electric separations, mixing and conveying, heating, cooling, adsorption, absorption, extraction, ion exchange, distillation, evaporation, sublimation, refrigeration, crystallization, and drying;
- Process design, construction, operation, control, and development with modeling, costing, simulation, optimization, process safety, pollution, energy integration, waste management, and reuse, circular economy, renewable energy.

Therefore, an ISO standard about quantities and units in these fields is truly needed. Besides it, some additional chapters in the IUPAC Green Book [3] would also be welcome.

### 3.1. Basic Chemical Engineering Principles and Unit Operations

Some basic quantities that are not included in other parts of ISO 80000 are presented in Table 1. Values of constants are taken from the SI brochure [11] and CODATA [13].

**Table 1.** Basic chemical engineering quantities.

| Name | Symbol | Definition | Unit | Remarks |
|------|--------|-----------|------|---------|
| Avogadro constant | $N_A$, $L$ | $N_A = N/n$ | $mol^{-1}$ | $6.022\,141\,76 \times 10^{23}$ |
| Boltzmann constant | $k$, $k_B$ | | $J\,K^{-1}$ | $1.380\,649 \times 10^{-23}$ |
| Faraday constant | $F$ | $F = eN_A$ | $C\,mol^{-1}$ | $9.648\,533\,212 \times 10^4$ |
| Henry's law constant | $k_H$ | $k_{H,B} = (\delta f_B / \delta x_B)_{x_B = 0}$ | Pa | |
| Planck constant | $h$ | $h = E/f$ | $J\,s$ | $6.626\,070\,15 \times 10^{-34}$ |
| Stefan-Boltzmann constant | $\sigma$ | $M_e = \sigma T^4$ | $W\,m^2\,K^{-4}$ | $5.670\,374\,419 \times 10^{-8}$ |
| 2nd virial coefficient 3rd virial coefficient | $B$ $C$ | $pV_m = RT\,(1 + B/V_m + C/V_m^2 + \dots)$ | $m^3\,mol^{-1}$ $m^6\,mol^{-2}$ | $pV_m = RT\,(1 + B_p\,p + C_p\,p^2 + \dots)$ |
| Coefficient of mass transfer | $k_c$ | $k_c = j_n / \Delta c_A$ | $m\,s^{-1}$ | |
| Specific surface area | $s$ | $S = A/m$ | $m^2\,kg^{-1}$ | |
| Logarithmic-mean temperature difference | $\Delta T_{lm}$ | $\Delta T_{lm} = (\Delta T_2 - \Delta T_1)/\ln(\Delta T_2/\Delta T_1)$ | K | LMTD |

Symbols—name, unit: $A$—area, $m^2$; $c$—concentration, $mol/m^3$; $e$—proton charge, C, $f$—fugacity: Pa, $j_n$—amount(-of-substance) flow, $mol/(m^2\,s)$; $M_e$—radiant exitance $W\,m^{-2}$, $N$—number of entities, 1; $n$—amount(-of-substance), mol; $p$—pressure, Pa; $R$—molar gas constant; $J/(mol\,K)$; $T$—thermodynamic temperature, K; $V_m$—molar volume, $m^3/mol$; $x$—amount(-of-substance) fraction, 1.

### 3.2. Chemical Reaction Engineering

Chemical reaction is the heart of chemical engineering activity—reactants are flowing into a reactor where they react, and products flow out of the reactor. Therefore, amount flow rates must be discussed first. The symbol $F$ is used in English literature [14], $\dot{n}$ Chemical Reaction Engineering in German one [15]. Mass flow rate, $q_m$ (kg/s), and volume flow rate, $q_V$ (m$^3$/s), are defined in ISO 80000-4-30.2 and 4-31, but amount-of-substance (shorter "amount" with the unit mol) flow rate is not; it is not defined in the Green Book [3], either. By analogy, q$_n$ (mol/s) could be used (Table 2). ISO 80000-4 defined mass flow, $j_m$, too; therefore, amount flow, $j_n$, is also included in the list.

**Table 2.** Chemical reaction engineering quantities.

| Name | Symbol | Definition | Unit | Remarks |
|---|---|---|---|---|
| Amount flow | $j_n$ | $j_n = cv$ | mol m$^{-2}$ s$^{-1}$ | |
| Amount flow rate | $q_n$ | $q_n = \int\int j_n \cdot e_n\, dA$ | mol s$^{-1}$ | $e_n$—normal vector |
| (Fractional) conversion | $X_B$ | $X_B = (n_B - n_{B0})/n_{B0} = 1 - c_B/c_{B0}$ | 1 | $dX_B = dc_B/c_{B0}$ |
| Selectivity | $\sigma_P$ | $\sigma_P = dc_P/(dc_P + dc_S)$ | 1 | |
| -(Fractional) yield | $\varphi$ $\Phi$ | $\varphi = dc_P/(-dc_A)$ $\Phi = c_{Pf}/(c_{A0} - c_{Af})$ | 1 1 | Instantaneous Overall, f—final |
| Rate of conversion | $\omega$ | $\omega = d\xi/dt$ | mol s$^{-1}$ | |
| Specific rate of conversion | $r_m$ | $r_m = (1/m)(dn_i/dt)$ | mol kg$^{-1}$ s$^{-1}$ | |
| Areic rate of conv. | $r_A$ | $r_A = (1/A)(dn_i/dt)$ | mol m$^{-2}$ s$^{-1}$ | |
| Volumic rate of conversion | $r_V$ | $r_V = (1/V)(dn_i/dt)$ | mol m$^{-3}$ s$^{-1}$ | V—reactor volume |
| Rate of reaction | $r_c$ r$_P$ | $r_c = (1/\nu_P)(dc_i/dt)$ r$_P = (1/RT)(dp_i/dt)$ | mol m$^{-3}$ s$^{-1}$ mol m$^{-3}$ s$^{-1}$ | For liquids For ideal gases |
| Rate constant | $k$ | $r = k\Pi_B c_B{}^{mB}$ | (m$^3$/mol)$^{m-1}$ s$^{-1}$ | $m$—order of reaction |
| Residence time distribution, RTD | $E$ | $\int_0^\infty E\, dt = 1$ | 1 | Age distribution at reactor exit |
| Space-time | $\tau$ | $\tau = V_r/q_{V,F}$ | s | F—feed |
| Pace-velocity | $s$ | $s = 1/\tau$ | s$^{-1}$ | |
| Recycle ratio | $R$ | $R = q_{V,r}/q_{V,f}$ | 1 | r—recycled, f—final |

Symbols—name, unit: $A$—area, m$^2$; $c$—concentration, mol/m$^3$; $n$—amount (-of-substance), mol; $p$—pressure, Pa; $t$—time, s; $v$—velocity, m/s; $\nu$—stoichiometric number, 1.

Conversion is the next quantity to be defined. It is often called fractional conversion (Umsatzgrad). The ISO 80000-9 and the Green Book cite extent of reaction, $\xi$ (mol), and the Green Book also rate of (absolute) conversion, $\dot{\xi} = d\xi/dt$ (mol/s). In American textbooks, the symbols $X_A$, or $x_A$, or $f_A$ are used for conversion of a reactant A, while $U_A$ (Umsatz) is used in German ones. $X_A$ is adopted here. Subscripts A, B, C, etc., are used for reactants, and P, R, S, etc., for reaction products. Selectivity, $\sigma_P$ is the amount ratio of desired product P to all products S formed. The definition in Table 2 is appropriate for reactors with constant volume. For selectivity calculation of a batch reactor, amounts of product P and reactant A are used $\sigma_P = n_P/(n_{A0} - n_A)$. For continuous reactors amount flows are needed, $\sigma_P = q_{n,P}/(q_{n,A0} - q_{n,A})$. Yield (Ausbeute) is the amount ratio of desired product P to reactant A fed. It can be instantaneous, $\varphi$, or overall, $\phi$. Yield is always the selectivity times the conversion, $\varphi_P = \sigma_P X_A$.

The name "rate of reaction" shall be used with constant volume fluids or with ideal gases only. In other cases, the rate of conversion for any species i is proposed to be used; the "specific rate of conversion" is applied in cases of solid in fluid-solid systems. The "areic rate of conversion" is suitable for interfacial surfaces in two-fluid systems and in

the surface of solid catalysts in gas-solid systems. The "volumic rate of conversion" is based on the volume of a reactor, not the volume of a fluid; it could also be named "rate of production", but this name is used in the case of selectivity, $q_{n,\text{B}} = \phi\, q_{n,\text{A0}} = \sigma_\text{B} X_\text{A}\, q_{n,\text{A0}}$.

Equilibrium constants are described in ISO 80000-9; space-time, space velocity, and yield are not. Space-time, $\tau$ (s), is the time required to fill a reactor volume with its volume flow rate of feed at specified conditions. Space velocity, s (s$^{-1}$), is the space-time reciprocal. Recycle ratio, $R$ (1), is the quotient of the volume flow rate returned to the reactor entrance and the one leaving the system.

Only a few quantities from reaction engineering are presented in Table 1. Single and multiple (series or parallel) reactions, elementary and nonelementary are known, and the number of molecules (molecularity with different orders of reaction) can differ and influence the rate equation. Temperature and pressure effects can vary, and the reaction can be exothermal or endothermal. In addition, we know different types of reactors—batch, plug flow, mixed flow, recycle ones. Flow patterns and contacting can be ideal or non-ideal; in the last case, dispersion, convection, or earliness of mixing must be accounted for. Finally, fluid-fluid (liquid or gas), fluid-solid, catalytic, and various biochemical (enzyme or microbial) reactors exist—heat and mass transfer become important in these cases, too. It will be difficult to standardize all of the quantities used in one standard. Especially so because process control, economics, and optimization influence the design of reactors.

Regarding axial dispersion, the dispersion coefficient, $D$ (m$^2$/s), mean time of a passage, $\bar{t}$ (s), and variance, $\sigma^2$, are important quantities; the probability distribution, statistics, and uncertainties are described in the Green Book [3] (pp. 151, 152). In the case of catalytic systems, the rate of conversion equations from Table 2 can be used; they can be based on the volume of voids in the reactor, mass or volume of catalyst pellets, catalyst surface area, or total reactor volume; activity of a catalyst, $a$ (1), may also be important. For heterogeneous reactions with two or more phases, the standard could contain some other quantities such as interfacial area density, $a$ (m$^2$/m$^3$), effectiveness factor ($\varepsilon$ or $\eta$, 1), mass transfer coefficient of the gas film, $\beta$, or $k_\text{g}$, or liquid film, $k_\text{l}$ (m/s), Henry's constant, $H$ (Pa m$^3$ mol$^{-1}$), Thiele modulus, $M_\text{T}$ (also $h_\text{T}$, or $\phi$ in German literature), Wagner or Weisz modulus, $M_\text{W}$, and Hatta modulus, $M_\text{H}$ (the unit 1 for all of them).

### 3.3. Other Unit Operations

Unit operations are numerous and differ very much from one another. Let us take distillation as an example. It is normal to write amount flow rates (mol/s) with a symbol of a flow rate name—F for feed flow rate, $D$ for distillate flow rate, S for side-stream flow rate, $V$ for vapor flow rate, etc. Correctly, $q_n$ could be used as a quantity symbol with a subscript denoting different flow rates, $q_{n,\text{F}}$, $q_{n,\text{D}}$, $q_{n,\text{S}}$, and $q_{n,\text{V}}$ in this case. The second disrespect of ISO 80000 rules is the name "duty" for the heat flow rate (W), e.g., condenser duty, reboiler duty, while their symbol, $\dot{Q}$ is in accordance with the ISO one. In addition, void fraction or even "voidage", $\varepsilon$ is not well defined—volume fraction of voids is the right name, and $\varphi$ the right symbol. Some other proposals for quantities of unit operations and their symbols are presented in Table 3.

**Table 3.** Quantity names, symbols, and units in separation units.

| Name | Symbol | Definition | Unit | Remarks |
|---|---|---|---|---|
| Amount flow | $j_n$ | $j_n = q_n/A$ | mol m$^{-2}$ s$^{-1}$ | |
| External reflux ratio | $R$ | $R = q_{n,\,N+1}/q_D$ | 1 | $q_{N+1}/V_N = R/(1+R)$ |
| Vapor-liquid equilibrium ratio | $K_i$ | $K_i = x_i/y_i$ | 1 | |
| Relative volatility | $\alpha_{ij}$ | $\alpha_{ij} = K_i/K_j$ | 1 | |
| Fugacity coefficient | $\phi_i$ | $\phi_i = f_i/p$ | 1 | $\phi_i = 1$ for ideal gas |
| Volume fraction of voids | $\varphi_v$ | $\varphi_v = V_v/V_{tot}$ | 1 | |

**Table 3.** *Cont.*

| Name | Symbol | Definition | Unit | Remarks |
|------|--------|-----------|------|---------|
| Efficiency of batch experiment | $\eta_b$ | $\eta_b = 1 - e^{-ktb}$ | 1 | $t_b$—batch mixing time |
| Efficiency of a continuous process | $\eta_c$ | $\eta_c = k\theta/(1 + k\theta)$ | 1 | $\theta$—total liquid residence time |

*3.4. Process Development and Design*

Process development data, which can be internal or external, process evaluation that includes capacity determination, and economics, process optimization, and decision making, are important. Table 4 presents the most frequent quantities in process engineering optimization, using mathematics and economics. Statistics is well covered in ISO standards; therefore, it will not be regarded here. Economics, on the other side, is not standardized, and often acronyms are used instead of symbols; it also lacks international coordination [16].

**Table 4.** Chemical and process engineering design economics.

| Name | Symbol | Definition | Unit | Remarks |
|------|--------|-----------|------|---------|
| Cost | $C$ | | EUR, USD, … | Cost index |
| Investment | $I$ | | €, $, … | Fixed capital |
| Interest rate | $i$ | | % | $V_p$—present value |
| Future value | $V_f$ | $V_f = V_p (1 + i)^N$ | 1 (% = $10^{-2}$) | $N$—number of years |
| Revenue, net sales | $R, S_n$ | $S_n = S_g - O_s$ | €, $, … | $S_g$—gross sales |
| Turnover ratio<br>Capital ratio | $r_{to}$<br>$r_c$ | $r_{to} = S_g/I$<br>$r_c = I/S_g$ | 1<br>1 | Reciprocals |
| Production rate | $q_m$ | $q_m = m/t$ | kg/s, t/a | Capacity dependent |
| Operating expenses | $O$ | $O = O_d + O_i$ | €, $, … | Direct + indirect expe. |
| Depreciation | $D$ | $D = I/N$ | €/a, $/a, … | With no salvage value |
| Gross income | $P_g$ | $P_g = R - O - D$ | €/a, $/a, … | Gross profit |
| Net income | $P_n$ | $P_n = Pg (1 - \tau)$ | €/a, $/a, … | Net profit, $\tau$—tax rate |
| Income tax | $T$ | $T = \tau (R - O - D)$ | % | |
| Net profit after tax | $P_n$ | $P_n = Pg (1 - \tau)$ | €/a, $/a, … | Net income |
| Cash flow rate | $q_c$ | $q_c = P_n + D$ | €/a, $/a, … | |
| Return on investment | $R_{oi}$ | $R_{oi} = P/I \times 100$ | % | Internal rate of return |
| Payout time | $t_{po}$ | $t_{po} = I/P_n$ | a | Payout period, years |

The most used cost indices are Marshall and Swift (M&S, since 1926), Chemical Engineering (CE, since 1958), and Nelson-Farrar (since 1946) ones. Capital investment includes equipment cost, instrumentation, piping, insulation, electrical, and engineering costs without any contingency; contingency is about 15%–20% of capital investment—when added to capital investment, the battery-limits capital investment is obtained. Working capital includes the fund for wages and salaries, purchase of raw materials, supplies, etc.

Operating expense is the sum of expenses for the processing of a product plus general, administrative, and selling expenses. They can be grouped into direct, indirect, and product expenses; direct expenses are raw materials, utilities, labor, maintenance, supervision, payroll charges, operating supplies, clothing and laundry, technical service, royalties, and environmental control. Indirect expenses include depreciation and plant indirect costs. Total manufacturing expense is adding packaging, loading, and shipping expenses to the operating expense. Revenues are the net sales received from selling a product to a customer. The value added to the product is the difference between the raw material expenses and the selling price of that product.

The time value of money is diminishing because of inflation. Interest rate includes the expectation that the borrowed capital should earn. The present value of money, $V_p$, is

lower than the future value, $V_f$. When a company loans money, a charge is made for the use of borrowed funds—the interest rate includes inflation expectation, the borrower's cost, and his desired profit. The cost of capital is what it costs the company to borrow money from all sources (loans, bonds, stocks); it is expressed as an interest rate.

Besides the term depreciation, quantity amortization is often used—there is a slight difference between them. If the period of life is known exactly, the annual expense is called amortization. If this time is estimated, it is called depreciation.

The rate of return and its variations are known by various names, e.g., internal rate of return, the interest rate of return, discounted cash flow rate of return.

### 3.5. Environmental Quantities, Units, and Symbols

Sustainable development with its three pillars (environmental, societal, economic ones) is gaining importance, and so is the Paris agreement with the 17 sustainable development goals (SDGs). The most problematic is the climate crisis caused by greenhouse gas (GHG) emissions with global warming. Table 5 presents some of the most important quantities in this area, starting with GHG emissions and climate change and ending with pollution.

**Table 5.** Environmental quantities with symbols and units.

| Name | Symbol | Definition | Unit | Remarks |
|---|---|---|---|---|
| Amount fraction of $CO_2$ equivalent | $x(CO_{2,eq})$ | $x_B = n_B / \sum_i ni$ | µmol/mol | In atmosphere |
| Emissions coefficient of electricity | $E_e(CO_{2,eq})$ | $E_e = m/W$ | kg/(kW h) | Not factor |
| Emission coefficient of travel | $E_l$ | $E_l = m/l$ | g/km | Various forms |
| Carbon footprint per user | $F_c$ | $F_c = m/t$ | t/a | Per person, . . . |
| Ecological footprint | $F_e$ | $F_e = A_{eq}$ | ha | |
| Water footprint | $F_w$ | $F_w = V/t$ | $m^3/a$ | |
| Amount fraction of air pollution | $x(SO_2)$ | $x = n_{SO2}/\Sigma n$ | nmol/mol | |
| Mass concentration of particulate matter pollutants, $d \leq (2.5, 10)$ µm | $\gamma_{PM2.5}$ $\gamma_{PM10}$ | $\gamma = m_{PM}/V$ | $µg/m^3$ | In air |
| Number concentration, e.g., microplastics | $C$ | $C = N/V$ | $m^{-3}$ | In lake, ocean |
| Mass concentration, heavy metal | $\gamma(Hg)$ | $\gamma = m_{Hg}/V$ | µg/L | In water |
| Mass fraction, heavy metal | $w(Pb)$ | $w = m_{Pb}/\Sigma m$ | mg/kg | In soil |
| Waste generation per capita | $q_m$ | $q_m = m/t$ | kg/a | Mass flow rate |
| Mass fraction of waste recycled | $w_r$ | $w_r = m_r/m_w$ | 1, % | Not recycling rate |

GHGs contain, besides the water vapor, $H_2O$, the most dangerous gases: carbon dioxide, $CO_2$, methane, $CH_4$, nitrous oxide, $N_2O$, ozone, $O_3$, chlorofluorocarbons, CFCs, and hydrofluorocarbons, HFCs; they are recalculated into $CO_2$ equivalents. The emissions coefficient of electricity is also named "electricity-specific emission factor", but the term "factor" should be used when the two quantities (mass and electricity in our case) have the same dimension, its unit is 1; the term "coefficient" shall be used when the two quantities have different dimensions. In ecological footprint, the unit global hectare, with the symbol gha, is used; it is wrong—the area taken into account is global, but the unit is hectare— global area, $A_g$, in ha [17]. The literature is usually writing about their concentrations or amount/volume fractions in ppm (parts per million) as a unit. The quantity is not concentration ($mol/m^3$) but rather amount fraction (µmol/mol). The units, ppm, ppb (part per billion), etc., are not recommended by IUPAC; therefore, amount (of substance) fraction with the symbol $x$ and unit µmol/mol, or nmol/mol, respectively, are used.

GHGs originate from the burning of fossil fuels in transportation, energy production, industry, residential areas, fermentation of waste, and agriculture. The $CO_{2, eq}$ emissions

can be expressed in different ways, e.g., as mass per energy produced (kg/(kW h), or mg/J), mass per volume of fuel (mg/L), mass per distance traveled (g/km), mass of $CH_4$ per agricultural area released or absorbed (kg/ha). They can be calculated per person, per company, per city, per country, or per world. No special names and symbols are available now. Many mistakes can be observed in statistical collections and in literature, e.g., by including the $CO_2$ formula or the words "per person" or "per capita" into the units.

In Table 5, some tentative symbols for quantities and units regarding emissions are proposed by respecting the ISO 80000 rules. Using the proposed names, symbols, and units, different forms of traveling (car, train, plane, etc.) or different users of carbon footprint can be addressed and compared. Many other footprints have been developed (water, land, nitrogen, phosphorus, material, biodiversity, chemical, plastic, energy, etc.); the environmental footprint family, relating to the nine planetary boundaries and their connection with SDGs, is being developed [18].

Pollution of air, water, and soil with chemical substances, heavy metals, particulate matter, noise, electromagnetic radiations, etc., is the second major environmental problem of modern society. Accepted terminology and symbols can be used for them, but they are usually not applied. An international standard could improve their usage. Waste minimization, recycling, and circular economy are becoming more and more important. "Waste generation" (per capita) actually means an abbreviated name for "mass flow rate of waste generated". "Recycling rate" means mass flow rate of recycled material ($q_m$, kg/a) that is a different quantity; therefore, the name is substituted by the "recycling fraction" in %.

## 4. Conclusions

Chemical and process engineering, as well as environmental science and engineering, are not represented in the 13 parts of ISO 80000 standard on quantities and units. The chemical industry alone sales in 2019 (3.66 TEUR—trillion euros, $10^{12}$ EUR) reach 4.2% of the world's GDP (gross domestic product, 74.76 TEUR). It is very interdisciplinary and specialized and different from the manufacturing industry. It is also very important in the area of sustainable engineering, especially in environmental sustainability. Therefore, it deserves a special part in the standardization of quantities and units. The literature review has shown that there are many names of quantities and units that are not in accordance with the ISO 80000 rules. Even worse is the situation with quantity symbols and units—many symbols of quantities are not coherent with the international system ISQ, and many units do not respect the SI rules. Acronyms cannot be used as quantity symbols, and SI units may not be intermixed with quantity specifications.

This paper tried to discuss and propose some of the most important quantities in the areas of chemical and environmental engineering and process economics. Regarding the names and symbols selected, the rules accepted in the systems of ISQ and SI were tried to be obeyed. The choice of quantities is, of course, just an illustration of names and symbols to be included in the proposal. Health and safety, statistics, management, and quality were not discussed as standards for them exist, but this does not mean that a review of quantity names, units, and symbols in those areas is not needed. In addition, basic concepts of process modeling, simulation, synthesis, design, integration, and optimization were not included yet.

The area is too broad and complex for the definite selection of quantities, their names, and symbols, but every journey starts with a single step. The proposal with the list of quantities, their names, symbols, and units must be discussed in national and international associations as well as the International Organization for Standardization. Environmental and economic quantities could also be discussed as separate standards because their importance is broader than the area of chemical and process industries.

**Funding:** This research was not funded.

**Conflicts of Interest:** The author declares no conflict of interest.

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
