# Peer review of "Quantities and Units in Chemical and Environmental Engineering"

_standards, doi:10.3390/standards2010004_

Round 1

Reviewer 1 Report

Comments Standards    Paper P.Glavic

It is utterly important to draw the attention of academic scientists and industrialist to the need of respecting the correct quantities and units in engineering. We are ll awre of its importance, but often forget it.

The manuscript is therefore an important reminder and guide.

It is certainly worthy of publication in Standards.

A few minor comments however

  1. Line 8: I don't like etc., but prefer among others.
  2. Tables:

* What is the physical meaning of a unit "1"...Should it not be indicated as "-", meaning no unit?

*Table 1: Log-mean T     the unit is K

*Table 1: Remarks   while 1,2,3,6 and 7 are values of the constants, it is difficult to understand 6, since symbols are not defined (in fact, will there be a nomenclature ?)

*Table 2: Remarks are explanation. What is meant by n - amount (of sub.) ?

*Table 3: again my question with unit = 1 (also in the text, e.g. Line 182)

*Table 5: what is meant by "not factor" and "not gha" ?

  1. Rest of the manuscript is O.K.

Minor revisions required.

Author Response

Please see the attachnent.

Reviewer 2 Report

Dear Author:

Thank you for your contribution, which builds on your existing work and research. The article is written in a clear manner and reflects the urgent need to introduce a uniform naming of units and quantities in environmental and chemical engineering for all parties involved. Moving into a new, more global level of environmental protection requires a uniform view on single units.

Please accept the following suggestions as a mean to clarify some parts of the work:

  • 2 mentions ISO 80000 and its somehow updated 13 Parts, with exception of parts 15÷17. However, line 68 mentions also Part 14. Please clarify the purpose and position of this part, as it is not mentioned in the text as such.
  • For easier orientation in the text, I suggest sorting the items in tables 1÷5 alphabetically.
  • Line 196÷228 – line spacing
  • 5 – Ecological footprint – My suggestion is to place the asterisk next to the gha*, and subsequently add the explanation directly under the tab. I found it easier for units that are not as commonly used or yet known, to be defined as close as possible to the given Table
  • In the section Methods you mentioned that „. the most important quantities have been selected according to the importance and frequency of their usage. “It would be beneficial for the readers if you develop on how you determined the importance and the frequency of the units.
